# Quality Teaching: Finding the Factors That Foster Student Performance in Junior High School Classrooms

**Vasiliki Polymeropoulou** [1,2,*] **and Angeliki Lazaridou** [1]

1   Department of Primary Education, University of Thessaly, 382 21 Volos, Greece; alazarid@uth.gr
2   School of Education, Frederick University, Limassol 3080, Cyprus
*   Correspondence: vpolymero@gmail.com

**Abstract:** In this study, we examined the characteristics of secondary school teachers that are effective at promoting student performance. Using a multilevel analysis design, we examined teachers' instruction using the effective factors in the Dynamic Model of Educational Effectiveness (D.M.E.E.). The research involved 21 teachers and 697 students. Written tests in Modern Greek were administered to the student sample at both the beginning and the end of the school year 2016–2017. An observer assessed teacher factors through three different instruments, and a student questionnaire was also used to evaluate teacher effectiveness. The data showed the effects that teachers in the sample had on the learning development of their students and highlighted effective teaching skills and weaknesses. Implications for practice to promote teacher effectiveness are drawn.

**Keywords:** teacher effectiveness; secondary schools; Dynamic Model of Educational Effectiveness; quality in education; student learning; multilevel regression modelling

## 1. Introduction

Researchers' interest in educational effectiveness increased significantly after two seminal studies by Coleman [1] and Jencks [2]. The findings of these studies highlighted the catalytic effect of students' backgrounds on their learning development but almost ignored the teaching factors as contributing to their improvement. Since then, many effectiveness studies have revealed the importance of teachers in the learning development of their students at rates that range from 15% to 25% [3–9]. Additionally, several studies conducted in the developing world have shown that teachers' influence on student performance ranges from 55% [10] to 60% [11,12]. These studies indicate that education is a significant lever for development, especially in a developing country. Therefore, an emphasis on improving the quality of teaching is indisputable as it may lead to the enhancement of students' learning outcomes. However, what does effective teaching consist of? Many effectiveness studies note that what teachers actually do in the classroom matters most for student improvement [13,14]. Although, there is no agreement about what constitutes effective teaching as many instructional factors have been highlighted by researchers (see Table 1).

**Table 1.** Instructional factors mediating effective teaching.

| Authors | Instructional Factors |
|---|---|
| Tucher & Stronge [15] | • Showing interest, respect, high expectations for all students, lesson preparation time, reflection<br>• Effective classroom management<br>• Use of a variety of teaching strategies and activities<br>• Course presentation in an understandable way<br>• Monitoring of students' learning<br>• Providing support and seeking the development of all students' abilities |
| Van de Grift [16] | • Ability of teachers to teach learning strategies to students, such as metacognition, problem solving, etc.<br>• Ability of teachers to adapt their teaching based on the needs of their students |
| Day et al. [17] | • Effective classroom management<br>• Educational approach: connecting the lesson with pre-existing knowledge, explaining, assigning of tasks, etc., using teacher-centered and/or student-centered approaches<br>• Creating supportive, positive, and inclusive learning environments for all students |
| Borich [18] | • Accuracy and clarity of the course presentation<br>• Variety<br>• Student involvement<br>• Degree of success of students<br>• Utilization of students' ideas by the teacher<br>• Lesson structure<br>• Questioning technique<br>• Encouraging student participation<br>• Creating interactive relationships between the teacher and students |
| Hattie [19] | • Connection of students' previous knowledge with new information<br>• Supportive learning climate (error is a means of learning)<br>• Ability to solve problems<br>• Monitoring students' learning progress and feedback<br>• Deep belief that all students can learn<br>• Awareness of the class/school context and adaptation of their teaching accordingly<br>• Teachers as agents of change |
| Raufelder et al. [20] | • Quality of the relationship between teacher and student (appreciation, personal interest, sympathy)<br>• Teaching experience (motivation, providing comprehensible teaching and variety of teaching)<br>• Characteristics of the teacher (empathy, self-confidence and humor) |

Charalampous and Praetorius [21] argue that the multiplicity of desirable teaching skills attests to the complexity of effective teaching and highlights the need for teachers and researchers to agree on which factors are to be considered most important. However, it is necessary to take into account the influence exerted each time on the teacher by the context in order to shape the appropriate teaching skills. For instance, recent research highlights the importance of other teaching skills related to diversity [22] and/or anti-racism education [23].

Efforts have been made in the last few decades to integrate teacher effectiveness factors with findings from school effectiveness research in order to develop theoretical models (e.g.,

Carroll and Comprehensive Model). In this notion, the Dynamic Model of Educational Effectiveness (D.M.E.E.) was developed by Creemers and Kyriakides [11].

## 2. The Dynamic Model of Educational Effectiveness at the Teacher Level

The dynamic model is multilevel and refers to the most important factors that operate at four levels: student, classroom/teacher, school, and system. The classroom level has the most important effect on student improvement, as the other levels have direct or indirect effects on students' learning outcomes [24]. The eight factors for teacher level that consistently have been shown to be associated with student outcomes [25] are shown in Table 2, with a short description of each:

**Table 2.** Factors of the D.M.E.E. associated with teacher effectiveness.

| Factor | Description |
|---|---|
| Orientation | Refers to teacher behavior that provides the students with opportunities to identify the reason(s) for activities that are presented in a lesson or a series of lessons, and/or activities that involve students in the identification of reason(s) for specific tasks. |
| Structuring | Teachers actively present materials and structure by:<br>• Beginning with an overview and/or review of objectives<br>• Outlining the content to be covered and signaling transitions between lesson parts.<br>• Calling attention to main ideas<br>• Reviewing main ideas at the end |
| Questioning | Refers to teacher skills in:<br>• Raising different types of questions (i.e., process and product) at appropriate difficulty level<br>• Giving time for students to respond<br>• Dealing with student responses |
| Teaching modelling | Refers to teachers' activities that encourage students to use or develop their own strategies in order to solve different types of problems. |
| Application | Refers to teachers' activities that provide students with practice and application opportunities. |
| Classroom as a learning environment | Include five components:<br>• Teacher–student interaction<br>• Student–student interaction<br>• Students' treatment by the teacher<br>• Competition between students<br>• Classroom disruption |
| Time Management | Refers to teacher's ability to:<br>• Prioritize academic instruction and allocate available time to curriculum-related activities<br>• Maximize student engagement |
| Assessment | Refers to teacher's ability to:<br>• Use appropriate techniques to collect data on students' knowledge and skills<br>• Analyze data in order to identify students' needs<br>• Report assessment results to students and parents<br>• Evaluate their own practices |

Each factor is measured in terms of the following five dimensions: frequency, focus, stage, quality, and differentiation [26–29]. These dimensions help us describe the presence

and quality of a factor. More specifically: *Frequency* is the only quantitative way of measuring the application and operation of each factor. However, examining only the frequency of activities related to a factor is not sufficient [11]. So, the other dimensions (focus, stage, quality, and differentiation) examine qualitative aspects of the factors. *Focus* is measured along two dimensions: the purpose/purposes with which each activity is carried out, and whether each activity is specific. Activities associated with a factor can be measured by considering the *stage* at which they take place, the reason being that the factors need to take place over a long period so as to ensure that they have a continuous direct or indirect effect on students' learning. *Quality* refers to the qualitative features of each factor, such as the level of student understanding of the materials taught and the degree of student involvement in the learning process [24,30]. *Differentiation* is the extent to which teaching activities are adapted to the needs of each group of students [31,32].

## 3. Studies on the Dynamic Model and Its Relation with the Dynamic Integrated Approach

When the dynamic model was first developed in 2008, it was used to evaluate the effectiveness of teachers on the dimension of quality. A number of studies were conducted with this orientation, resulting in the validation of its main components and highlighting the factors that assess teachers' effectiveness. Table 3 refers to these studies and the main focus of each.

**Table 3.** Studies that used the dynamic model.

| Study | Main Focus |
| --- | --- |
| Kyriakides & Creemers [33] | Teacher and school effectiveness in different subjects (i.e., mathematics, language and religious education) and different learning domains (cognitive and affective). |
| Kyriakides & Creemers [34] | The impact of teaching factors on achievement in mathematics and language of Cypriot students at the end of pre-primary school. |
| Kyriakides et al. [29] | Synthesis of 167 studies on the impact of generic teaching skills on student achievement. |
| Panayiotou et al. [35] | The validity of the dynamic model, by investigating the impact of teaching factors on student achievement in mathematics and science. |
| Azkiyah et al. [36] | The effects of two intervention programs on teaching quality by considering the impact of teaching factors on student achievement in mathematics. |
| Azigwe et al. [10] | Observation and student questionnaire data to measure the impact of teaching factors on the mathematical achievement of primary students in Ghana. |
| Ioannou [37] | The impact of teacher factors on slow learners' outcomes in language. |
| Kokkinou & Kyriakides [38] | Whether teachers exhibit the same generic teaching skills when they teach in different classrooms. |
| Kyriakides et al. [39] | Integrating generic and content-specific teaching practices when exploring teaching quality in primary physical education. |
| Kyriakides et al. [40] | The impact of teacher behavior for promoting students' cognitive and metacognitive skills. |
| Dimosthenous et al. [41] | The short- and long-term effects of the home learning environment and teacher factors included in the dynamic model on student achievement in mathematics. |

Four conclusions can be drawn from the above description:

(1) The theoretical background of the dynamic model for teacher effectiveness has been validated extensively,

(2) The teaching factors and their dimensions contribute significantly to student achievement, emphasizing the importance of the factors for the evaluation of cognitive learning outcomes in mathematics, language, science, religious studies, and physical education. (It should be noted that these studies mainly concerned primary school teachers.)

(3) It is possible to measure the quantitative as well as the qualitative characteristics of each factor.

(4) The "Teacher" level of the dynamic model is based on multiple learning theories to define effective teaching.

Effective teaching factors based on the dynamic model can be arranged along a continuum that starts with direct teaching, such as structuring and questioning, and extends to using new learning theories and approaches such as constructivism including factors, e.g., orientation and modelling. As a result, teachers' instructional behavior can be classified into five stages (see Table 4), structured in a developmental order [42].

**Table 4.** The five stages of teaching skills.

| Stages |
| --- |
| 1st Stage: Basic elements of direct teaching |
| 2nd Stage: Putting aspects of quality in direct teaching and touching on active teaching |
| 3rd Stage: Acquiring quality in active/direct teaching |
| 4th Stage: Differentiation of teaching |
| 5th Stage: Achieving quality and differentiation in teaching using different approaches |

A teacher's placement on the developmental stages continuum can be a criterion for determining (a) the type of feedback that the teacher is to receive about the quality of their teaching as well as (b) the support they need in order to advance to a higher stage.

This scalable differentiation of effective teaching has been explored by Kyriakides et al. [43] and Antoniou et al. [26], and has resulted in the creation of the *Dynamic Integrated Approach* (D.I.A.) to professional development for teachers [44]. In this way, the dynamic model is not limited to evaluating the effectiveness of teachers but links it to their professional development.

## 4. Significance of the Study

This research is expected to have significant implications in the field of educational effectiveness research. The findings offer validated information to those who are searching for studies on secondary teachers' effectiveness. Through the lens of the dynamic model, the effectiveness of instructional teaching skills was examined, evidenced during a course on Modern Greek that was being delivered to junior high school students. We chose to focus on this level of schooling because all other studies conducted with the dynamic model have focused on primary education, and also because in the Greek system for education, secondary and primary schools are organized and operate quite differently. One of the contributing factors, for example, is that secondary school teachers must have a university degree with specialization in one or more school curriculum subjects but, unlike elementary school teachers, they are not required to have any preparation in pedagogy. Other contributing factors include, of course, the usual disparities in students' physiological and intellectual development, as well as differences in social maturity that correlate with differences in the dynamics of interpersonal relationships among students, as well as with their teachers [45]. In addition, there are the usual adolescence-related changes that trigger intense socio-emotional instability and, in turn, affect students' learning immensely [46,47].

Another important aspect of this study is the fact that it focused on the subject of language whereas previous studies with the dynamic model have focused mainly on mathematics [43,48]. A final reason for conducting our study is that few studies using

the dynamic model have been focused on the effectiveness of the teacher in the Greek educational context [35,49]. This research gap is being filled by the present research.

Based on the above, considerations, we used the following question to scaffold our study: "What characteristics of secondary teachers' instruction, as measured in a language course, are effective in promoting student performance?"

## 5. Method

### 5.1. Larger Context

The study was carried out in central Greece, where schools are 44% urban and 56% rural.

### 5.2. Research Sample

#### 5.2.1. Pilot Study

After obtaining the requisite permissions to conduct the research, a preliminary meeting with the principals and the teachers took place to inform them about the purpose and procedures of the study. The selection of schools and teachers for both the pilot research and the main research was based on three criteria. The first was the socio-economic level of the students; schools of both urban and regional areas were represented in the sample. Secondly, due to the increased demands of the research (teaching observations) but also the fact that one observer was conducting it, the participating teachers should be approached as soon as possible. So, the initial choice of the researcher is to exclude schools that are located at a great distance from the center of the prefecture, which was the base of the researcher. The selection of these schools and teachers would be made only in case the sample would not be satisfactory. Third, parental permission for the participating students was necessary. It was positive that after the information meeting given to the schools, the teachers showed a positive response to their participation in the research. The schools that were informed decided whether they would participate in the pilot phase or in the main study.

Before the main research was carried out, a pilot study was conducted, involving two language counsellors, four language teachers, a sample of schools ($n = 5$) and teachers ($n = 6$). These schools were excluded from the main study. The pilot diagnostic test was administered to 92 third graders of junior high while the final pilot test was taken by 128 first graders of senior high school, as they were the most suitable to complete them due to their completion of the curriculum of the third grade in junior high school. The test that was developed for this purpose reflected the requirements of the Greek curriculum for junior high. Before pilot testing the (before and after) written tests, feedback was sought from two language counsellors and four language teachers to verify that the written tests covered the taught material, were of varying difficulty and corresponded to the students' learning age. At the beginning of the school year, the pilot application of the two written tests was carried out in order to predict ambiguities. All the tests were administered and corrected by the researcher in order to increase the reliability of the measurement. After considering the results of the pilot tests, the final diagnostic and final written tests were formed.

#### 5.2.2. Main Study

The sample of the main study consisted of teachers in secondary education who taught modern Greek language in the 3rd grade of junior high school and the students of these classes. Secondary education in the Greek education system consists of 3 grades of junior high (called Gymnasium) and 3 grades of senior high (called Lyceum). In total, twenty-one secondary school teachers and their students ($n = 697$) participated in this main study for one school year. Three teachers expressed reluctance and one teacher was excluded (even though they wanted to participate) because parents' consent could not be obtained. It is worth noting that some teachers were very positive about participating, because they saw this study as an opportunity for them to receive feedback on their teaching skills.

More specifically, the total number of teachers who taught this specific cognitive subject in the prefecture was 37. Among them, 21 teachers consented to participate in the

main study (56.8%) and 6 participated in the pilot study. Background information—despite not being taken into account in the research—nevertheless gives the profile of participated teachers. The number of female teachers participating in the study was 18 (86%) and the number of male teachers was 3 (18%). As regards the years of experience of the teacher sample, 10 of them had 10 to 20 years and 11 had over 20 years. In terms of their scientific qualifications, only two teachers had a postgraduate degree specializing in educational sciences. Of the participating teachers, 9 (43%) taught the specific subject in one class per school, while 12 (57%) taught two classes.

The participating students, on the other hand, amounted to 697 (50.4%) out of a total of 1382 students in the third grade of junior secondary school of the specific prefecture. In terms of gender, the percentage of participating boys was 49.9% and that of girls was 50.1%. No statistically significant differences between the participating students and the total number of students in terms of gender ($\chi^2(1) = 0.651$, $p = 0.420$) were found. In terms of the geographical area of the students that participated in the study, 48.1% came from an urban area and 51.8% from a regional area. Finally, it is worth noting that a necessary condition for a student to be included in the final sample of the study was to have participated in two written tests and the student questionnaire. As a result, students who, for some reason, did not participate in a certain phase were excluded. However, this loss for the present study was minimal, so there were no variations in the final sample.

*5.3. Variables and Data Collection*

5.3.1. Dependent Variables: Achievement in Modern Greek Language

Students' achievements in the Modern Greek class were measured at the beginning and end of the school year. The test that was developed for this purpose reflected the requirements of the Greek curriculum for junior high.

In both measures, the Extended Logistic Model of Rasch [50] was used to analyze the data. The analysis revealed that the infit mean squares and the outfit mean squares of each scale were near one and the values of the infit t scores and the outfit t scores were approximately zero. Therefore, each analysis revealed that there was a good fit to the model [51]. Thus, for each student, it was possible to generate two different scores for their achievement in language—one at the beginning and one at the end of the 3rd grade.

5.3.2. Explanatory Variables at the Student Level

Two explanatory variables were also used in this study: aptitude and student background information. *Aptitude* refers to the degree to which a student is able to perform the learning task. For the purpose of this study, students' prior knowledge in modern Greek language was measured. In order to control for the effects of *student background factors*, a student questionnaire was used to collect data on the students' background. Four variables associated with the Social Economic Status (SES) of the students' background were sought: (a) the social status of the father's job, (b) the social status of the mother's job, (c) the economic status of the family and (d) the immigration background. Following the classification of occupations used by the Greek Ministry of Finance, it was possible to classify the parents' occupation into three groups with relatively similar sizes: (a) occupations held by working class, (b) occupations held by middle class, and (c) occupations held by upper middle class. The SES indicator was the average of the four standardized values.

5.3.3. Explanatory Variables at Classroom Level: Quality of Teaching

To measure the quality of teaching, four instruments were used: a student questionnaire, a high-inference observation instrument, and two low-inference observation instruments.

The students' questionnaire evaluated the eight factors and their dimensions of teacher effectiveness identified in the dynamic model. Specifically, students were asked to indicate the extent to which their teacher behaved in a certain way in their classroom, and a Likert-scale was used to collect data. For example, an item concerned with the stage dimension of the structuring factor asked students to indicate whether the teacher would explain at

the beginning of a new lesson how the lesson would relate to previous ones. Similarly, the following item "The teacher assigns to some pupils different exercise than to the rest of the pupils" examined how the differentiation dimension of the application factor was measured [52].

The High-Inference (HI) observation instrument evaluates all eight factors of the model with the five dimensions. The observer completes a Likert scale to indicate how often each teacher behavior was observed. For example, an item concerned with the frequency dimension of orientation asks the observe to indicate the extent to which the teacher spent time explaining the objectives of the lesson [29].

The first Low-Inference (LI 1) observation instrument evaluated the five factors of the model (orientation, structure, application, modeling of teaching, and questioning techniques). This instrument was designed in a way to collect more information about the quality dimensions of these five factors.

The second Low-Inference (LI 2) observation instrument assessed the learning environment of the classroom, as evident in the interactions between the teacher and students, the students with each other, the management of the classroom, and the management of teaching time. More specifically, the instrument is based on the Flanders system of interaction. A classification system of teacher behavior that is based on the way each factor of the dynamic model is measured was developed. For example, in order to measure the quality dimension of teacher behavior in dealing with misbehavior, the observer tried to identify if the teacher was using any strategy to deal with a disruption problem or not, if the use of any strategy had a long-lasting effect and finally if the misbehavior was solved but only temporally. The instrument also includes a classification system of student behavior in order to gather data about the three elements of the factor classroom as a learning environment and the management of time factor. All these instruments have been applied previously in a series of studies and their construct validity and reliability have been provided by the previous studies and tested using Structural Equation Modeling (SEM) techniques [30]. The three observation instruments are presented in a book written by Creemers and Kyriakides [11].

*5.4. Data Collection*

Written tests in Modern Greek were administered to the student sample at both the beginning and end of the school year in order to see the student's achievement in language.

To measure the teacher factors, observations by an independent observer and a student questionnaire were used. The classrooms' observations were conducted between November 2016 and April 2017 by the researcher who previously had been trained to use the three observation instruments. For each participating teacher who was teaching one course, three teaching observations were made; in the case of participating teachers who taught in two classes, two observations were made per class. In all, 79 teaching observations were conducted, of which 37 used LI 1 and 42 used the LI 2. The HI was used at the end of each observation. At the end of the school year, the students' questionnaire was issued, along with assessment of their socio-economic level and the quality of teachers' teaching.

## 6. Analysis of Data

Separate multilevel analyses were conducted with *MLwiN* [53], and the data were conceptualized as a two-level model—that is, students at the first level, and classrooms/teachers at the second level.

The first step in the analysis was to determine the variance at each level without the explanatory variables ("empty model"). This model contains random groups and random variation within groups. The dependent variable is the sum of a general average ($\beta_0$), a random effect at the classroom/teacher level ($U_{0j}$) and a random effect at the student level ($R_{ij}$).

$$Y_{ij} = \beta_0 + U_{0j} + R_{ij} \text{ (empty model)}$$

Here, $Y_{ij}$ expresses the results of student i at the end of the year, who was taught by teacher j. The random parts $U_{0j}$ and $R_{ij}$ are considered to have zero means and are

independent of each other. By using this model, the separation of data variability at two levels is achieved.

In the next phase of data analyses, the students' background factors (i.e., SES and prior achievement) and their aggregated scores at the classroom and school levels were added (Model 1). The decision to consider the aggregated effects of background factors was based on the findings of studies that investigated group composition effects. The findings revealed that aggregated scores have, in some countries, even stronger effects on final achievement than the individual factors. More specifically, two meta-analyses of studies investigating the effect of SES on student achievement revealed that when researchers used aggregated measures of SES, they usually reported much higher correlations between SES and academic achievement than when they used individual measures of SES [54,55]. It is also important to note that variables measuring background factors were standardized as Z scores with a mean of 0 and a standard deviation of 1.

$$Y_{ij} = \beta_0 + \beta_1 X_{1ij} + \beta_2 X_{2ij} + \beta_3 X_{3ij} + \ldots + U_{0j} + R_{ij} \text{ (Model 1)}$$

where $X_1$ = initial performance and $X_2$, $X_3$ ... = student characteristics that appeared to be related to learning outcomes at the end of the school year.

In the next phase, the effectiveness factors that were based on the classroom/teacher level were gradually added. More specifically, in Model 2, the factors were added one by one, in the order they appeared on each form. The following equation was applied to examine the effect of the *Orientation* factor on learning outcomes at the end of the year.

$$Y_{ij} = \beta_0 + \beta_1 X_{1ij} + \beta_2 X_{2ij} + \beta_3 X_{3ij} + \beta_4 X_{4ij} + \beta_5 X_{5ijk} + \beta_6 \text{ (Orientation)}_j + U_{0j} + R_{ij} \text{ (Model 2}\alpha)$$

This analytic procedure was applied to the data for each factor in order to investigate its particular effect.

Finally, in Model 3, all factors of teacher effectiveness were added, in order to check for the power of each factor when considered simultaneously. The independent variables were recorded as distances from the average (centered—grand mean) as z scores. Thus, each result expresses how much the dependent variable increases (or decreases, in the case of a negative sign) by any additional deviation in the independent variable [56].

## 7. Results

Appendix A Tables A1–A5 present the results of each multilevel analysis in relation to students' achievement levels in the Modern Greek class. In all four analyses, dispersion was statistically significant at every level. The findings from the "empty model" analysis reveal that 89% of the variance in language skill is at the student level, and 11% at the classroom/teacher level. Moreover, the "empty model" reveals that in the analysis of variance at each level, there is statistical significance ($p \leq 0.05$), which implies that multilevel analysis can be used to identify the explanatory variables that are associated with achievement levels in particular subjects.

The Model 1 analysis revealed, first of all, that two factors—namely, students' prior knowledge and SES—had a significant impact on the students' final performance. Not surprisingly, prior knowledge had the larger effect (Cohen's $\delta = 0.76$) compared to the effect of SES (Cohen's $\delta = 0.16$). The interpreted dispersion found at the student level, related to their prior knowledge and SES, maxed out at 43%. However, approximately 48% of the total variance remained unexplained at the student level. Second, the effect of both background factors at the student level (SES, prior knowledge) was found to be statistically significant ($p \leq 0.05$). Moreover, chi square showed a statistically significant change between Model 1 and Model 2 ($p \leq 0.001$), which means that there is a differential effect of SES and prior knowledge at the classroom/teacher level. Third, the results reveal that prior knowledge (i.e., aptitude) has the strongest effect in predicting student achievement at the end of the school year. This finding is in line with similar results from other effectiveness studies [57,58] and it showed the importance of this factor to student's achievement.

The next step was to add the teaching quality factors one by one, drawing on data from the three observation instruments and the student questionnaire. Thus, various forms of Model 2 arose. The suitability of each form of Model 2 was examined in comparison with Model 1 via the statistical criterion $\chi^2$ and differences between Model 1 and each form of Model 2 were evident ($p < 0.001$), which indicates that the measurement variables of the factors of a teacher have a significant effect on the students' learning outcomes. Finally, it shows that all the factors of the dynamic model have a significant effect on student learning outcomes except from the assessment

## 8. Discussion and Implications

This study confirmed that the teacher effectiveness factors referenced in the dynamic model are valid for assessing junior high school teachers, thus verifying the effectiveness of the model at the secondary level. In the following discussion, we compare the information generated through this research with understandings gained from previous investigations.

### 8.1. Effect of Teachers on Students' Learning Outcomes

The findings indicate that there is an effect of the teacher on student performance, which reaches 11%. This effect, however, is smaller than similar percentages obtained from other studies. For example, a study by Kyriakides & Creemers [39], which investigated the effect of the teacher on the learning development of kindergarten and elementary school students, produced a reading of 15%. Similarly, the research conducted by Christoforidou, Kyriakides, Antoniou, and Creemers [27] showed that the influence of primary school teachers on the final performance of their students is about 17%. A much larger study by Panayiotou, Kyriakides, Creemers, McMahon, Vanlaar, Pfeifer, Rekalidou, and Bren [35], involving six European countries (Belgium, Cyprus, Greece, Slovenia, Germany, and Ireland), showed that the effect of primary school teachers on students' cognitive development in mathematics was around 24%. Finally, in this connection, a more recent study by Azigwe, Kyriakides, Panayiotou, and Creemers [10] on the influence of the teacher on the learning development of students in Ghana showed a rate of 55%.

The comparatively smaller percentage of teachers' effectiveness may be attributed to specific features of Greek secondary education which are quite different from those at the elementary level. For instance, as indicated previously, secondary teachers in Greece do not receive adequate training in pedagogy and teaching methodologies during their preservice preparation, nor do they mitigate this inadequacy with professional development in their in-service years. As a result, they lack substantial knowledge in both pedagogical skills and teaching methodologies. Another explanation may lie in the management protocols for the Greek education system, especially at the secondary level, which contribute to frequent teacher transfers, fragmentation of curricula, and deficiencies in the teaching methods that teachers use. The knowledge-centered and mechanical character of the Greek secondary educational sphere has been pointed out in a report by the Quality Assurance Authority in Primary and Secondary Education [59], and these realities have been an ongoing struggle for a large number of teachers in secondary education. In concert, these factors may well be responsible for the smaller effect of teaching on student performance that we found. As Scheerens [60] has suggested, the smaller effect of the teacher on students' achievement in language studies, compared with their achievements in other subjects, is likely to contribute more to students' language abilities and skills compared to other factors, such as the students' family background.

### 8.2. Teaching Factors That Affect Students' Learning Improvement

It is essential to note that our findings are "mapping" the specific factors associated with effective teaching approaches in secondary education. Through this mapping, it is now possible to identify teachers' strengths and weaknesses and to direct them towards personalized professional development. As shown, seven of the eight factors of teacher effectiveness appeared to be significantly related to student development. The only factor

that did not appear to play a significant role was that of assessment. It seems that this factor was not in our teachers' repertoires. However, it is worth noting here that assessment was measured only through the students' questionnaire. It is possible that different effects of this factor may have occurred if the data at issue had been collected by teachers too. In fact, in the study by Christoforidou and Xirafidou [49], which was carried out with primary education teachers, the effect of assessment on the learning development of students was evident. It should be noted, however, that this study used a self-reported questionnaire. Another explanation for the lack in the present study of an effect related to the assessment factor may be related to the above-mentioned administrative structure for Greek education. In the Greek context, assessment is not treated as a significant pedagogical function that helps improve students and teachers alike. Rather, emphasis is placed on the summative test results of students [59]. Therefore, although research has shown that assessment of students by teachers is an important correlate of improving students and teachers [27,49,61,62] and a characteristic of their effectiveness [11,42,63,64], in the Greek context, this factor operates differently due to the specificities of the administrative system at the secondary level.

Finally, in regard to the issue at hand, according to Kyriakides, Creemers, Panayiotou, and Charalambous [24], it seems that assessment is analyzed in five developmental stages through the five dimensions (frequency, focus, stage, quality, and differentiation), starting with simpler evaluation techniques and ending with the most complex ones, which contain more qualitative characteristics of the dimensions. Therefore, a different empirical approach is needed to adequately investigate this factor.

Another significant finding of this study relates to the differentiation dimension. In our data, this dimension was not present in any factor, which suggests that the participating teachers did not take the needs of students into account during teaching. Interestingly, in the literature, the assessment and differentiation factors seem to be related to one another. According to Moon [65], assessment is essentially the first step in moving towards differentiating their teaching. This adds importance to our finding during instruction for students' needs. The teachers in our sample opted for a one-size-fits-all approach in their teaching [66], which runs counter to the view that contemporary classrooms are a mosaic of students with differentiated characteristics [67].

*8.3. The Effect of Students' Prior Knowledge*

From the data analysis, it emerged that the students' previous knowledge base had a greater impact on their final performance than their socio-economic background. This has also been found to be the case in other studies [52,57,58,68–71]. Related to this is the finding, in a number of studies [33,41,48,72], that points out the influence of previous teachers on students' prior knowledge and, in turn, points to the importance of teachers' long-term influence, whether they are effective or not. This long-term effect has been known since 1996, when Sanders and Rivers [73] claimed that a teacher's effects can last for at least two school years. Later, Rivers and Sanders [74] found a longer lasting effect, reaching four years.

## 9. Conclusions

Quality in teaching has been a core concern in modern schooling around the world. Identifying the factors that contribute to quality in teaching and teachers is a constant challenge for all education systems and will continue to draw the attention of educational researchers in the years ahead. In Greece, it is only in the last decade that discussion about the quality of education has surfaced. This interest has led to a number of studies that have resulted in valuable insights about effectiveness factors. One such contribution is the study reported here, which is part of a much larger effort to investigate the characteristics and impacts of secondary school teachers in Greece. Accordingly, we leveraged the dynamic model to extend this research thrust and have sharpened our understandings about the teaching skills that significantly affect student performance. Through the lens of the dynamic model, we have identified specific factors that are associated with quality instruction

and thereby have opened up avenues for improving teachers' effectiveness. We suggest that venturing down those paths can have numerous benefits in both the short and long run [75–77]. The provision of carefully targeted training opportunities to teachers—guided by the eight factors of the D.M.E.E. that relate to quality and effective instruction—has to be a priority at the policy level. In fact, a study by Kyriakides, Christoforidou, Panayiotou, and Creemers [78], which used the dynamic approach (D.I.A.) in teacher professional development programs, revealed that the D.I.A. had a significant impact on improving teachers' instructional skills by helping them to progress to more advanced levels. The present study has shown that Greek secondary school teachers could benefit from similar professional development programs, especially when focused on assessment and differentiation functions. This would enhance teachers' instructional efficiency and effectiveness on the one hand and students' achievement levels on the other.

**Author Contributions:** Conceptualization, V.P.; methodology, V.P., formal analysis, V.P.; resources, V.P.; writing—original draft preparation, V.P.; writing—review and editing, V.P. and A.L.; supervision, A.L. All authors have read and agreed to the published version of the manuscript.

**Funding:** This research received no external funding.

**Institutional Review Board Statement:** The study was conducted according to the guidelines of the Declaration of Helsinki, and approved by the Ethics Committee of Institute of Educational Policy (IEP), (162391/Δ2/03-10-2016).

**Informed Consent Statement:** Informed consent was obtained from all subjects involved in the study.

**Conflicts of Interest:** The authors declare no conflict of interest.

## Appendix A

**Table A1.** Calculation of parameters and typical errors for the analysis of students' language performance (students in classes) at the end of the 3rd grade (with the data obtained from the low-inference observation form that emphasizes the classroom environment).

| Factors | Model 0 | Model 1 | Model 2a | Model 2b | Model 2c | Model 3a | Model 3b | Model 3c |
|---|---|---|---|---|---|---|---|---|
| **Stable class (Intercept)** | −0.78 (0.13) +++ | −0.77 (0.11) +++ | −0.77 (0.11) +++ | −0.77 (0.10) +++ | −0.77 (0.11) +++ | −0.77 (0.11) +++ | −0.77 (0.10) +++ | −0.77 (0.10) +++ |
| *Student level* | | | | | | | | |
| Prior achievement | | 0.76 (0.03)+++ | 0.77 (0.03) +++ | 0.76 (0.03) +++ | 0.76 (0.03) +++ | 0.76 (0.03) +++ | 0.76 (0.03) +++ | −0.76 (0.03) +++ |
| SES | | 0.16 (0.06) +++ | 0.16 (0.06) +++ | 0.16 (0.06) +++ | 0.16 (0.06) +++ | 0.16 (0.06) +++ | 0.19 (0.06) +++ | 0.16 (0.06) +++ |
| *Class level* | | | | | | | | |
| Teacher–student interaction—*Quality* | | | 0.14 (0.11) + | | | 0.07 (0.03) +++ | | 0.21 (0.11) ++ |
| Management of time | | | | 0.06 (0.02) +++ | | | 0.50 (0.02) +++ | 0.05 (0.03) + |
| Teacher–student interaction—*Frequency* | | | | | 0.05 (0.03) + | | | 0.05 (0.04) |
| **Variance components** | | | | | | | | |
| Class | 10.84% | 8.53% | 7.98% | 7.00% | 7.98 % | 7.08% | 7.06% | 5.97% |
| Student | 89.16% | 48.32% | 48.32% | 48.32% | 48.32% | 48.32% | 48.32% | 48.30% |
| Interpreted | | 43.15% | 43.70% | 44.68% | 43.70% | 44.60% | 44.62% | 45.73% |
| **Significance test** | | | | | | | | |
| $\chi^2$ | 2846.07 | 2428.53 | 2426.94 | 2422.91 | 2426.88 | 2424.36 | 2420.54 | 2419.31 |
| Reduction | | 417.54 | 1.59 | 5.62 | 1.65 | 2.58 | 2.37 | 3.6 |
| Degrees of freedom | | 2 | 1 | 1 | 1 | 1 | 1 | 1 |
| *p*-value | | 0.001 | 0.001 | 0.001 | 0.001 | 0.001 | 0.001 | 0.001 |

Note: For Model 1, the reduction was calculated in relation to the deviation of Model 0; For Models 2a–2c, the reduction was calculated in relation to the deviation of Model 1; For Model 3a, the reduction was calculated in relation to the deviation of Model 2a; For Model 3b, the reduction was calculated in relation to the deviation of Model 2b; For Model 3c, the reduction was calculated in relation to the deviation of Model 2b, since it explains most of the discrepancy in relation to the other models in its category (i.e., 2a–2c); + Variable significant at level 0.10; ++ Variable significant at level 0.05; +++ Variable significant at level 0.001.

**Table A2.** Calculation of parameters and typical errors for the analysis of students' language performance (students in classes) at the end of the 3rd grade (with the data obtained from the low-inference observation form with the five factors).

| Factors | Model 0 | Model 1 | Model 2a | Model 2b | Model 2c | Model 2d | Model 2e | Model 3 |
|---|---|---|---|---|---|---|---|---|
| **Stable class (Intercept)** | −0.78 (0.13) +++ | −0.77 (0.11) +++ | −0.77 (0.10) +++ | −0.78 (0.10) +++ | −0.77 (0.10) +++ | −0.77 (0.09) +++ | −0.77 (0.11) +++ | −0.78 (0.09) +++ |
| *Student level* | | | | | | | | |
| Prior achievement | | 0.76 (0.03)+++ | 0.77 (0.03) +++ | 0.77 (0.03) +++ | 0.77 (0.03) +++ | 0.77 (0.03) +++ | 0.76 (0.03) +++ | 0.78 (0.03) +++ |
| SES | | 0.16 (0.06) +++ | 0.16 (0.06) +++ | 0.16 (0.06) +++ | 0.16 (0.06) +++ | 0.16 (0.06) +++ | 0.16 (0.06) +++ | 0.15 (0.06) +++ |
| *Class level* | | | | | | | | |
| Orientation—*Frequency* | | | 0.05 (0.02) +++ | | | | | 0.03 (0.02) |
| Structuring—*Stage* | | | | 0.33 (0.15) +++ | | | | 0.14 (0.16) |
| Structuring—*Focus* | | | | | 0.26 (0.14) ++ | | | 0.02 (0.14) |
| Structuring—*Quality* | | | | | | 0.72 (0.19) +++ | | 0.57 (0.22) +++ |
| Application—*Frequency* | | | | | | | 0.012 (0.008) ++ | 0.012 (0.007) ++ |
| **Variance components** | | | | | | | | |
| Class | 10.84% | 8.53% | 7.30% | 7.08% | 7.35% | 4.93% | 7.95% | 3.87% |
| Student | 89.16% | 48.32% | 48.32% | 48.32% | 48.32% | 48.32% | 48.32% | 48.30% |
| Interpreted | | 43.15% | 44.38% | 44.60% | 44.33% | 46.75% | 43.73% | 47.83% |
| **Significance test** | | | | | | | | |
| $\chi^2$ | 2846.07 | 2428.53 | 2424.65 | 2423.78 | 2425.28 | 2416.32 | 2426.97 | 2411.48 |
| Reduction | | 417.54 | 3.88 | 4.75 | 3.25 | 12.21 | 1.56 | 4.84 |
| Degrees of freedom | | 2 | 1 | 1 | 1 | 1 | 1 | 1 |
| *p*-value | | 0.001 | 0.001 | 0.001 | 0.001 | 0.001 | 0.001 | 0.001 |

Note: For Model 1 the reduction was calculated in relation to the deviation of Model 0; For Models 2a–2e, the reduction was calculated in relation to the deviation of Model 1; For Model 3, the reduction was calculated in relation to the deviation of Model 2d, since it explains most of the discrepancy in relation to the other models in its category (i.e., 2a–2e); [++] Variable significant at level 0.05; [+++] Variable significant at level 0.001.

**Table A3.** Calculation of parameters and typical errors for the analysis of students' performance in language (students in classes) at the end of the 3rd grade (with the data obtained from the students' questionnaire).

| Factors | Model 0 | Model 1 | Model 2a | Model 2b | Model 2c | Model 2d | Model 2e | Model 2f |
|---|---|---|---|---|---|---|---|---|
| **Stable class (Intercept)** | −0.78 (0.13) +++ | −0.77 (0.11) +++ | −0.77 (0.11) +++ | −0.77 (0.11) +++ | −0.77 (0.11) +++ | −0.77 (0.10) +++ | −0.77 (0.11) +++ | −0.77 (0.11) +++ |
| *Student level* | | | | | | | | |
| Prior achievement | | 0.76 (0.03)+++ | 0.76 (0.03) +++ | 0.76 (0.04) +++ | 0.75 (0.03) +++ | 0.75 (0.03) +++ | 0.76 (0.04) +++ | 0.75 (0.04) +++ |
| SES | | 0.16 (0.06) +++ | 0.15 (0.06) +++ | 0.15 (0.06) +++ | 0.15 (0.06) +++ | 0.15 (0.06) +++ | 0.16 (0.06) +++ | 0.15 (0.06) +++ |
| *Class level* | | | | | | | | |
| Structuring | | | 0.18 (0.09) +++ | | | | | |
| Management of time | | | | 0.18 (0.10) ++ | | | | |
| Application | | | | | 0.25 (0.09) +++ | | | |
| Teaching modelling | | | | | | 0.17 (0.06) +++ | | |
| Orientation | | | | | | | 0.13 (0.09) + | |
| Questioning | | | | | | | | 0.24 (0.08) +++ |
| **Variance components** | | | | | | | | |
| Class | 10.84% | 8.53% | 7.90% | 8.09% | 7.79% | 7.38% | 8.47% | 7.93% |
| Student | 89.16% | 48.32% | 48.32% | 48.32% | 48.32% | 48.32% | 48.32% | 48.32% |
| Interpreted | | 43.15% | 43.78% | 43.59% | 43.89% | 44.30% | 43.21% | 43.75% |
| **Significance test** | | | | | | | | |
| $\chi^2$ | 2846.07 | 2428.53 | 2424.51 | 2425.29 | 2421.17 | 2417.88 | 2426.35 | 2419.98 |
| Reduction | | 417.54 | 4.02 | 3.24 | 7.36 | 10.65 | 2.18 | 8.55 |
| Degrees of freedom | | 2 | 1 | 1 | 1 | 1 | 1 | 1 |
| *p*-value | | 0.001 | 0.001 | 0.001 | 0.001 | 0.001 | 0.001 | 0.001 |

Note: For Model 1, the reduction was calculated in relation to the deviation of Model 0; For Models 2a–2f, the reduction was calculated in relation to the deviation of Model 1; For Model 3a, the reduction was calculated in relation to the deviation of Model 2a; For Model 3b, the reduction was calculated in relation to the deviation of Model 2b; For Model 3c, the reduction was calculated in relation to the deviation of Model 2c; For Model 3d, the reduction was calculated in relation to the deviation of Model 2d; For Model 3e, the reduction was calculated in relation to the deviation of Model 2d, since it explains most of the discrepancy in relation to the other models in its category (i.e., 2a–2f); + Variable significant at level 0.10; ++ Variable significant at level 0.05; +++ Variable significant at level 0.001.

**Table A4.** (Continued) Calculation of parameters and typical errors for the analysis of students' performance in language (students in classes) at the end of the 3rd grade (with the data obtained from the students' questionnaire).

| Factors | Model 3a | Model 3b | Model 3c | Model 3d | Model 3e |
|---|---|---|---|---|---|
| **Stable class (Intercept)** | −0.77 (0.11) +++ | −0.77 (0.11) +++ | −0.77 (0.11) +++ | −0.77 (0.10) +++ | −0.77 (0.10) +++ |
| *Student level* | | | | | |
| Prior achievement | 0.76 (0.04) +++ | 0.76 (0.04) +++ | 0.75 (0.04) +++ | 0.76 (0.03) +++ | 0.74 (0.04) +++ |
| SES | 0.49 (0.30) ++ | 0.49 (0.30) ++ | 0.35 (0.26)+ | 0.28 (0.20) + | 0.15 (0.06) +++ |
| *Class level* | | | | | |
| Structuring | 0.22 (0.10) +++ | | | | 0.03 (0.10) |
| Management of time | | 0.19 (0.11) ++ | | | 0.01 (0.11) |
| Application | | | 0.28 (0.10) +++ | | 0.14 (0.10) + |
| Teaching modelling | | | | 0.19 (0.07) +++ | 0.08 (0.08) |
| Orientation | | | | | 0.08 (0.09) |
| Questioning | | | | | 0.13 (0.10) + |
| **Variance components** | | | | | |
| Class | 6.95% | 7.06% | 6.76% | 7.36% | 7.19% |
| Student | 48.32% | 48.32% | 48.32% | 48.32% | 48.30% |
| Interpreted | 44.73% | 44.62% | 44.92% | 44.32% | 44.51% |
| **Significance test** | | | | | |
| $\chi^2$ | 2422.25 | 2422.16 | 2418.55 | 2415.44 | 2411.20 |
| Reduction | 2.26 | 3.13 | 2.62 | 2.44 | 6.68 |
| Degrees of freedom | 1 | 1 | 1 | 1 | 1 |
| *p*-value | 0.001 | 0.001 | 0.001 | 0.001 | 0.001 |

Note: For Model 1, the reduction was calculated in relation to the deviation of Model 0; For Models 2a–2f, the reduction was calculated in relation to the deviation of Model 1; For Model 3a, the reduction was calculated in relation to the deviation of Model 2a; For Model 3b, the reduction was calculated in relation to the deviation of Model 2b; For Model 3c, the reduction was calculated in relation to the deviation of Model 2c; For Model 3d, the reduction was calculated in relation to the deviation of Model 2d; For Model 3e, the reduction was calculated in relation to the deviation of Model 2d, since it explains most of the discrepancy in relation to the other models in its category (i.e., 2a–2f); + Variable significant at level 0.10; ++ Variable significant at level 0.05; +++ Variable significant at level 0.001.

**Table A5.** Calculation of parameters and typical errors for the analysis of language performance of students (students in classes) at the end of the 3rd grade (with the data obtained from the high-inference observation form, calculated—aggregated at the level of class).

| Factors | Model 0 | Model 1 | Model 2a | Model 2b | Model 2c | Model 2d | Model 2e | Model 2f | Model 3a | Model 3b |
|---|---|---|---|---|---|---|---|---|---|---|
| **Stable class (Intercept)** | −0.78 (0.13) +++ | −0.77 (0.11) +++ | −0.77 (0.11) +++ | −0.78 (0.10) +++ | −0.77 (0.11) +++ | −0.78 (0.11) +++ | −0.77 (0.11) +++ | −0.77 (0.11) +++ | −0.77(0.11) +++ | −0.79 (0.10) +++ |
| *Student level* | | | | | | | | | | |
| Prior achievement | | 0.76 (0.03) +++ | 0.76 (0.03) +++ | 0.76 (0.03) +++ | 0.76 (0.03) +++ | 0.76 (0.03) +++ | 0.76 (0.03) +++ | 0.76 (0.03) +++ | 0.76 (0.04) +++ | 0.76 (0.03) +++ |
| SES | | 0.16 (0.06) +++ | 0.16 (0.06) +++ | 0.16 (0.06) +++ | 0.16 (0.06) +++ | 0.16 (0.06) +++ | 0.16 (0.06) +++ | 0.16 (0.06) +++ | 0.25 (0.19) + | 0.16 (0.06) +++ |
| *Class level* | | | | | | | | | | |
| Structuring | | | 0.15 (0.12) + | | | | | | 0.17 (0.12) + | 0.04 (0.19) |
| Orientation | | | | 0.27 (0.13) +++ | | | | | | 0.37 (0.24) + |
| Management of time | | | | | 0.32 (0.23) + | | | | | 0.04 (0.33) |
| Student–student interaction | | | | | | 0.34 (0.19) +++ | | | | 0.14 (0.20) |
| Classroom disorder | | | | | | | 0.17 (0.12) + | | | 0.46 (0.30) + |
| Learning environment: Supportive | | | | | | | | 0.24 (0.19) + | | 0.62 (0.49) + |
| **Variance components** | | | | | | | | | | |
| Class | 10.84% | 8.53% | 7.95% | 7.30% | 7.93% | 7.65% | 8.06% | 8.12% | 6.95% | 6.32% |
| Student | 89.16% | 48.32% | 48.32% | 48.32% | 48.32% | 48.32% | 48.32% | 48.32% | 48.32% | 48.30% |
| Interpreted | | 43.15% | 43.73% | 44.38% | 43.75% | 44.03% | 43.62% | 43.56% | 44.73% | 45.38% |
| **Significance test** | | | | | | | | | | |
| $\chi^2$ | 2846.07 | 2428.53 | 2426.90 | 2424.75 | 2426.65 | 2425.55 | 2426.72 | 2426.97 | 2423.67 | 2421.81 |
| Reduction | | 417.54 | 1.63 | 3.78 | 1.88 | 2.98 | 1.81 | 1.56 | 0.23 | 2.94 |
| Degrees of freedom | | 2 | 1 | 1 | 1 | 1 | 1 | 1 | 1 | 2 |
| *p*-value | | 0.001 | 0.001 | 0.001 | 0.001 | 0.001 | 0.001 | 0.001 | 0.001 | 0.001 |

Note: For Model 1, the reduction was calculated in relation to the deviation of Model 0; For Models 2a–2f, the reduction was calculated in relation to the deviation of Model 1; For Model 3a, the reduction was calculated in relation to the deviation of Model 2a; For Model 3b, the reduction was calculated in relation to the deviation of Model 2b since it explains most of the discrepancy in relation to the other models in its category (i.e., 2a–2f); + Variable significant at level 0.10; +++ Variable significant at level 0.001.

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
