# Peer review of "Quality Teaching: Finding the Factors That Foster Student Performance in Junior High School Classrooms"

_education, doi:10.3390/educsci12050327_

Round 1

Reviewer 1 Report

The overall manuscript contains relevant information for existing literature. One aspect that you can focus on is consistency in terminology and punctuation. There are spaces missing (or an extra space is added). You can also make improvements in terms of concise writing (some sections are wordy). This would positively contribute to the quality.

Author Response

Cover Letter

Thank you for giving us the opportunity to make corrections and changes to the manuscript, following the reviewer’s very helpful comments.

Below we present in detail all the interventions we have made in order to respond to the specific remarks as best as possible.

1st Reviewer

Changes

p.1.9 search out = incorrect. Alternative: examined.

corrected

p.1.13/14 This sentence is incomplete.

Add details

p.1.27

corrected

p.1.29 clearly = redundant.

corrected

p.2.Table1 Do not center the text. In addition, the table contains inconsistencies in the use of periods, commas, and capital letters. Please make this consistent (also throughout your manuscript; see e.g., other Tables). Please re-check for double spaces. Three or more authors can be immediately listed as first author, followed by et al. Apply this throughout your manuscript.

All suggested corrections have been made

p.2.39 next step example of the lack of spaces. Please go over your manuscript because I found this issue often. From this point onward, I am not going to point this out anymore.

corrected

p.2.40 not limited = not only limited?

corrected

p.3.54-60 Merge the two paragraphs

corrected

p.4.73 You can use the abbreviation here again.

corrected

p.5.91/92 these should be em dashes rather than en dashes.

The punctuation wasn’t right so I made the correction

p.5.Table 4 The column with the stages is redundant. You can make one column with stage and description.

I made the change of the structure of table 4

p.5.118-123 This section is fragmented. Can you create more coherence? In addition, the section p.5.121-123 is redundant (it is a repetition).

All suggested corrections have been made

p.6.RQ Why are you phrasing it as could be ? It sounds doubtful and at this point in your manuscript, that should not be the case.

corrected

p.6.Table5

I removed the table and incorporated the information into the text. Please see p.7, 192-208

p.6.138 En dash is incorrect here.

corrected

p.6.145 Place the n in italics. Also insert spaces.

corrected

p.6.footnote This is not how you present a footnote. The number needs to be in superscript.

corrected

p.7.189-194 Present the five factors with (a), (b), (c), to guide the reader through your manuscript.

corrected

p.8.206 ..this criterion did not come into play What do you mean with this?

clarification was made Please see p.7, 208-212

p.8.223 I cannot figure out if you present information in superscript or not (U0j).

the explanations of the symbols are given below. Please see p.9, 305-307

p.9.280 Avoid back-to-back brackets.

corrected

p.9.288/289 You can just state that your aspects of teaching had a significant effect on quality. The Table is redundant/unnecessary.

the table has been removed and the information included in the text, please see p. 10, 367-368

p.10.330 Avoid construction with more important (i.e., perhaps more important ). Your reasoning should indicate its essence (not the use of intensifiers and/or subjective labeling).

corrected

p.10.333 Refer to a table with a capital letter (Table 5 instead of table 5).

corrected

p.11.350-353 This is in complete. What does that imply?

the necessary remarks were made

p.11.359 of little concern Can you come up with a more formal/academic description?

the expression was removed

p.11.362 strongly Avoid this intensifier. This does not add anything relevant to your message.

 was removed

p.11.364 Another important Phrase this differently.

corrected

p.11.369-372 They did not increase it; they found a longer lasting effect.

corrected

p.11.377 or so = informal. Can you come up with a formal alternative?

was removed

p.Appendices

corrected

p.References

corrected

Reviewer 2 Report

See attached file.

Author Response

Cover Letter

Thank you for giving us the opportunity to make corrections and changes to the manuscript, following the reviewer’s very helpful comments.

Below we present in detail all the interventions we have made in order to respond to the specific remarks as best as possible.

2rd Reviewer

Changes

Revise the introduction/literature review to include a more comprehensive discussion of teacher effectiveness studies.

The necessary changes have been made to make it appear that I am basically referring to instructional teaching skills. Please see p.1, 23-36,p.2 42-50

Table 5 has a layout that is difficult to understand

Due to the difficulty of reading the table 5, the information was chosen to be embedded within the text

Please see p.7, 192-208

I am struggling with the first sentence or two in 5.4.4. At the top of the Data collection section (p. 8, lines 204-206), there is a sentence or so that outlines inclusion criteria; students were included in the study only if they had taken a written achievement test at the beginning and end of the year. However, this criterion was said to have “not come into play.” Is this the written achievement test proposed here different than the “written test” that is described just after this point?

The necessary changes were made to make the process more understandable. Also, the specific information was transferred to the section of research sample of main study as it was considered to fit more. Please see p.7, 208-212

Throughout the paper, words are combined in a way that they should not be: “studieshighlighted” on p. 1, line 23), “ratesthat” (p. 1, line 27), and others.

Due to different software that I used these failures occur and the words appear together. I used another computer so that this problem would not occur

Reviewer 3 Report

Dear authors

The topic presented has a great potential in the field of educational research. Results should contribute to educational quality improvement. However, it is essential that they arise from a scientific investigation that, in addition to being well implemented, all its phases must be well described: theoretical basis, problems, objectives, methodology, data collection instruments, participants.

In general, the article fails to describe important phases of the investigation. Data collection instruments are not (well) described. This makes it difficult to justify analysis categories. It would be crucial to present the questions (observable variables) that originated latent variables (categories) in order to understand on what data the analyzed dimensions are based. This is a negative aspect.

Additionally, participant selection process is not (well) described. It is important to justify the number of participants. This is a negative aspect.

Therefore, I consider that weaknesses pointed out must be improved to ensure data reliability. 

Author Response

Cover Letter

Thank you for giving us the opportunity to make corrections and changes to the manuscript, following the reviewer’s very helpful comments.

Below we present in detail all the interventions we have made in order to respond to the specific remarks as best as possible.

 3rd Reviewer

Changes

Data collection instruments are not (well) described. This makes it difficult to justify analysis categories.

Additions were made in order to make the observational instruments more detailed. Please see p.8, 242-249, 256-280

participant selection process is not (well) described. It is important to justify the number of participants.

The text was modified and the selection process of the participants was made more detailed. Please see p.6, 150-162, p. 7, 184-191.

Round 2

Reviewer 3 Report

Dear authors

I consider that the changes made to the work took into account all my recommendations and contributed to the overall article quality improvement. Minor errors must be corrected (attached document). Regards.

Author Response

Cover Letter

Reviewer 3

I would like to express my sincerely feelings for your accurate markings.

  1. Pag. 5 indifferentiationin (I suggest indifferentiation in)

Answer:

the change to differentiation in

  1. and Antoniou [47[50] and Antoniou,

Answer:

 I made the appropriate change so the reader could understand that I mentioned two different studies. So. I change it to ‘explored by Kyriakides et al.  Creemers, and Antoniou [50] and Antoniou et al. Creemers, and Kyriakides [5]’,

  1. pag. 6

to 92 3rd graders (I suggest 923rd)

128 1st (I suggest 1281st)

 Answer:

The proposed changes have been made

  1. pag. 7

[x2(1) =0,651 (I suggest 2)

Answer:

The proposed changes have been made